# CT Perfusion as a Predictor of the Final Infarct Volume in Patients with Tandem Occlusion

**DOI:** 10.3390/jpm13020342

**Published:** 2023-02-16

**Authors:** Giordano Lacidogna, Francesca Pitocchi, Alfredo Paolo Mascolo, Federico Marrama, Federica D’Agostino, Alessandro Rocco, Francesco Mori, Ilaria Maestrini, Federico Sabuzi, Armando Cavallo, Daniele Morosetti, Francesco Garaci, Francesca Di Giuliano, Roberto Floris, Fabrizio Sallustio, Marina Diomedi, Valerio Da Ros

**Affiliations:** 1Stroke Center, Department of Systems Medicine, University Hospital of Rome “Tor Vergata”, Viale Oxford 81, 00133 Rome, Italy; 2Diagnostic Imaging Unit, Department of Biomedicine and Prevention, University of Rome Tor Vergata, Viale Oxford 81, 00133 Rome, Italy; 3Interventional Radiology Unit, Department of Biomedicine and Prevention, University Hospital of Rome “Tor Vergata”, Viale Oxford 81, 00133 Rome, Italy

**Keywords:** stroke, CT perfusion, tandem occlusion, mechanical thrombectomy

## Abstract

Background: CT perfusion (CTP) is used in patients with anterior circulation acute ischemic stroke (AIS) for predicting the final infarct volume (FIV). Tandem occlusion (TO), involving both intracranial large vessels and the ipsilateral cervical internal carotid artery could generate hemodynamic changes altering perfusion parameters. Our aim is to evaluate the accuracy of CTP in the prediction of the FIV in TOs. Methods: consecutive patients with AIS due to middle cerebral artery occlusion, referred to a tertiary stroke center between March 2019 and January 2021, with an automated CTP and successful recanalization (mTICI = 2b − 3) after endovascular treatment were retrospectively included in the tandem group (TG) or in the control group (CG). Patients with parenchymal hematoma type 2, according to ECASS II classification of hemorrhagic transformations, were excluded in a secondary analysis. Demographic, clinical, radiological, time intervals, safety, and outcome measures were collected. Results: among 319 patients analyzed, a comparison between the TG (N = 22) and CG (n = 37) revealed similar cerebral blood flow (CBF) > 30% (29.50 ± 32.33 vs. 15.76 ± 20.93 *p* = 0.18) and FIV (54.67 ± 65.73 vs. 55.14 ± 64.64 *p* = 0.875). Predicted ischemic core (PIC) and FIV correlated in both TG (tau = 0.761, *p* < 0.001) and CG (tau = 0.315, *p* = 0.029). The Bland–Altmann plot showed agreement between PIC and FIV for both groups, mainly in the secondary analysis. Conclusion: automated CTP could represent a good predictor of FIV in patients with AIS due to TO.

## 1. Introduction

In acute ischemic stroke (AIS) due to large vessel occlusion (LVO) of the anterior brain circulation, endovascular treatment (EVT) is considered effective both alone or in combination with intravenous rTPA administration in the first 6 h [1,2,3,4,5,6]. After 6 h, the combined use of CT perfusion (CTP) and multiphasic CT angiography (mCTA) enables the selection of patients with LVO who may still benefit from EVT by discriminating among definitive ischemic core volume and still salvageable brain tissue (penumbra) [7,8,9,10,11,12,13,14], finally predicting functional outcome [15].

The CTP technique is essentially based on the dynamic detection of time–density curves (TDC) for a selected brain area generated after bolus injection of contrast medium. In the post-processing of CTP images, it is necessary to select an input artery as a reference to determine the start time of the first circulation and then analyze the cerebral parenchyma blood perfusion. Then, different mathematical models are applied to calculate the perfusion parameters including cerebral blood flow (CBF), cerebral blood volume (CBV), mean transit time (MTT), time to peak, and time to maximum (Tmax). Each parameter is displayed as a parametric map of the brain with a color scale representing the values [16,17,18].

The automated CTP system RAPID (iSchemaView, Inc., Menlo Park, CA, USA) was implemented in clinical trials in order to provide an accurate prediction of ischemic core volumes by estimating CBF reduction < 30% with respect to the contralateral hemisphere, and the surrounding penumbra, detected by prolonged time-to-maximum peak (T_max_ > 6 s). Nowadays it is an essential tool for treatment decision making outside the 6 h time window [7,8,16,17,18].

Tandem occlusions (TOs), consisting of a simultaneous extracranial internal carotid artery (ICA) and intracranial LVO, represent about 15% of all LVO strokes [19,20,21]. There are relatively few patients with TOs included in the major randomized controlled trials of EVT and a lack of robust data prevents the delineation of definitive treatment recommendations [22,23,24,25].

From a CT perfusion perspective, it is not completely clear if the concomitant occlusion of the extracranial ICA can influence the pharmacokinetic deconvolution model needed to create the density–time curve and the colorimetric maps [26,27,28,29].

Just a few works, with conflicting results, explore the accuracy of perfusion parameters according to different arterial input function (AIF) locations, such as when a stenosis is present before the intracranial occlusion, increasing the uncertainty in CTP accuracy in discriminating ischemic core [30,31,32].

In clinical settings, both Haussen et al. and Albers et al. [9,24] found CTP reliable enough in predicting ischemic core volume in ipsilateral extracranial steno-occlusive disease, when appropriate delay correction was applied to CTP analysis, in spite of the coexistence of delay and dispersion effects due to an extracranial occlusion.

However, we noticed that in both studies the Bland–Altmann plot, employed for evaluating the degree of agreement between predicted infarct core (PIC) and final infarct volume (FIV), showed a positional bias. Namely, PIC and FIV mean differences directly correlated with PIC and FIV means, as if the accuracy in predicting FIV was dependent on final ischemic core volumes [9,24,32,33].

In this study, we aimed to evaluate the possible differences between the PIC, estimated with automated CTP, and the FIV, in patients with AIS due to TO that underwent successful recanalization with EVT. Then, we tried to disentangle if biological factors which could finally interfere with FIV estimation, such as parenchymal hemorrhages with mass effect (PH2) and successful but partial recanalization (i.e., modified thrombolysis in cerebral infarction [mTICI] 2b-2c), could influence the prediction accuracy.

## 2. Methods

### 2.1. Clinical Data and Procedural Times

A single center and prospectively collected database of patients with AIS treated with EVT from a tertiary care academic institution was retrospectively reviewed. Between March 2019 and January 2021, all consecutive patients with AIS due to middle cerebral artery (MCA) occlusion with a pretreatment CTP with automated RAPID software post-processing and who underwent EVT and achieved a successful recanalization were collected. According to the presence of a concomitant extracranial ICA occlusion, patients were classified into two groups: the tandem group (extracranial ICA + MCA) (TG) and the control group (MCA occlusion only) (CG). The MCA occlusion patients were furtherly divided into the proximal M1, distal M1 (beyond perforating artery emergency), and M2 segments. T occlusion were excluded from the analysis in order to avoid further uncertainty in perfusion patterns. Successful recanalization was defined as a final mTICI grade of 2b, 2c, or 3 [34]. Full recanalization was defined as an mTICI of 3. Baseline demographic data, medical history, pre-stroke modified Rankin scale (mRS) score, and baseline National Institute of Health Stroke Scale (NIHSS) were collected by a neurologist from the neurovascular unit at admission. All patients in the proper time window who received rTPA according to the most recent stroke guidelines were recorded, too. Due to the influence of time on the dynamic ischemic core development, onset to CTP, CTP to recanalization, and onset to recanalization times were also recorded. Due to the uncertain onset time, both “wake up stroke” and “unknown time of onset stroke” were excluded from the analysis regarding the “onset to recanalization” and the “onset to reperfusion” analysis.

### 2.2. Radiological Measures

The imaging protocol included in order: non-contrast CT (NCCT), CT angiography (CTA), and CT perfusion (CTP); these were performed using a 128-slice CT scanner (Revolution; General Electric Healthcare, Waukesha, WI, USA). NCCT parameters were: scan type: axial; gantry tilt: orbitomeatal plane; slice thickness: 2.5 mm/5 mm base/cerebrum; interval: 20 mm; kV: 120; mA: 120–300; rotation time: 1 s; and matrix: 512 × 512. To define the presence of TO occlusion (extracranial ICA stenosis > 90%) a bolus of 70 mL iodinated contrast agent was administered at a flow rate of 3.5–4 mL/s. The CTA was acquired in a volumetric way including the epiaortic vessels. The vessel occlusion was defined first on the mCTA and then confirmed on conventional angiography. The CTP protocol was the subsequent 50 mL of iodinated contrast agent (370 mgI/mL) that was power-injected at a rate of 5 mL/s followed by 50 mL saline bolus at 5 mL/s. The large detector array (8 cm) allowed evaluation of all the MCA territory. The protocol included 2 phases (scan type: axial-shuttle; slice thickness: 5 mm; kV: 80; mA: 200; rotation time: 0.5 s): The first phase started with a 5-second delay from contrast injection, and 22 acquisitions were repeated on the same 8 cm section (total scan duration: 60.6 s); the second phase was delayed 15 s and consisted of one acquisition on the same section (total scan duration: 15 s). Images were processed using a commercially available software tool (RAPID version 4.5.0). The total hypoperfused volume was defined by a >6 s delay for the maximum of the tissue residue function (T_max_ > 6 s). The ischemic core lesion was defined by a cerebral blood flow (CBF) reduction to <30% of the corresponding contralateral territory. The target mismatch profile was defined as a core ≤ 50 cm^3^, absolute mismatch ≥ 15 cm^3^, T_max_ > 10 s ≤ 100 cm^3^, and the mismatch ratio > 1.8, with processed parametric maps overlaid upon the source CTP data for review purposes. Furthermore, 2 neuroradiologists with at least 5 years of experience performed a case-by-case check of the AIF and VOF curves parameters necessary to create the density/time curves and the colorimetric maps through the deconvolution algorithm [26,27,28]. After EVT, an MRI within 24–36 h of symptom onset was performed in a subgroup of patients with a 1.5 T scanner system (Intera, Philips Medical System, Best, The Netherlands) using an 8-channel head coil; the MRI protocol included an axial T2-weighted fluid-attenuated inversion recovery (FLAIR) sequence (repetition time (TR) 6000 ms, echo time (TE) 120 ms, at 5 mm) and diffusion-weighted imaging (DWI) (TR 3023 ms, TE 89 ms, at 5 mm) acquired using a single-shot echo-planar sequence with fat suppression in the axial plane with 3 b values (0, 500 and 1000 s/mm^2^). In case of contraindications to perform the MRI (metallic implants, claustrophobia, pacemakers (although new protocols allow imaging in selected cases), MR-incompatible prosthetic heart valves, body weight (MRI tables have specific weight limitations)), a non-contrast CT was performed. The final infarct volume was measured using a dedicated computing platform (3D Slicer software) [35] by 2 neuroradiologists with 5 years of experience who outlined the ischemic core on ADC images (DWI images with b = 1000 s/mm^2^).

### 2.3. Procedural Details and Measures

In patients with airway impairment, general anesthesia was administered. The stability on common carotid artery was reached with a 6 cm diameter long sheath. After diagnostic arteriograms and the identification of target lesions, the operator decided between a proximal–distal or distal–proximal approach. The ICA and intracranial vessels were navigated with a guiding catheter and the EVT was performed using either direct aspiration thrombectomy or the “Solumbra” technique (combination of stent retriever and direct aspiration thrombectomy).

The clot burden score (CBS) and the collateral score (CS) were evaluated on the basis of the Tan et al. [36] study. CS was further subdivided in a dichotomous way (CS > 1 (Tan CS > 1) = good; CS < 1 (Tan CS < 1) = poor).

### 2.4. Outcome Measures

Functional outcome was recorded by means of mRS at 3 months both in an ordinal and dichotomous way. Namely, mRS scores between 0 and 2 were considered a good functional outcome (functional independence), scores of 3 and 4 were considered a poor functional outcome, and mRS scores of 5 and 6 were considered an unfavorable outcome. As per safety measures, hemorrhagic transformation (HT) was graded based on the ECASS II classification [37]. Symptomatic HT was defined as an intracerebral hemorrhage (ICH) with an increase in the NIHSS Score of ≥4 points. Since PH2 presumably altered the accuracy of the final infarct volume determination, all subjects who suffered a massive parenchymal hemorrhage in the 24–48 h after EVT were excluded from the secondary analysis. Since according to ECASS II classification, smaller hemorrhagic transformations should not significantly influence FIV evaluation, we retained such cases in the analysis. Reocclusion, vasospasm, and arterial embolism in another vascular territory were also recorded. Patients who experienced any of these procedural complications were excluded from the analysis.

### 2.5. Statistical Analysis

Continuous variables are expressed as means (±SD) or medians (IQR). Categorical variables are expressed as the sample absolute number (N) and proportions (%). Between-groups comparisons of continuous or ordinal variables were made with Student’s *t*-test or the Mann–Whitney U test, as appropriate. Categorical variables were compared by the χ^2^ or Fisher exact test, as appropriate. Correlation coefficients were calculated with non-parametric Kendall’s Tau-b statistic. The Bland–Altman plot was performed to evaluate the degree of agreement of measurements between predicted core infarct by CTP CBF < 30% volume and final infarct volume in patients with TO and those without [33]. Significance was set at *p* < 0.05. Statistical analyses were performed using SPSS^®^ Statistics 21 (IBM^®^, Armonk, NY, USA). The study was approved by the local ethical committee.

## 3. Results

Between March 2019 and January 2021, among 319 patients with AIS who underwent EVT, 59 patients met the inclusion criteria. A total of 22 (37%) were treated for concomitant extracranial ICA and MCA occlusion (TG) while 37 (63%) experienced an AIS exclusively due to a MCA occlusion (CG) (detailed description of patient selection in Appendix A). Groups’ baseline, clinical, interventional, and outcome measures are reported in Table 1 and Table 2. TG showed significantly higher median NIHSS (U = 544.000; *p* = 0.031) and lower CBS (U = 60.5; *p* < 0.001). In the TG a large artery occlusion etiology prevailed, while in the CG a cardioembolic one prevailed. In the CG, two patients with ICA dissection and one patient with aortic complicated plaque were included. The TG showed significant longer times from CTP to recanalization (mean 135.74 ± 48.51 vs. 107.49 ± 37.93; U = 534.00; *p* = 0.46) due to longer procedural times (67.68 ± 47.09 vs. 40.14 ± 26.95; U = 566.00; *p* = 0.013), and needed a higher median number of passages (2 vs. 1, U = 576.500; *p* = 0.004) to achieve successful recanalization. The “Solumbra” technique was significantly more employed in the TG than CG (χ^2^ = 9.394; *p* = 0.02). No significant differences were detected between groups for outcome variables (outcome data are reported in Appendix A for the whole sample and in Appendix A according to the hemorrhagic subtype in the Appendix A). PH2 in the 24–48 h after EVT was found in 4 patients (18.2%) in the TG and in 9 patients (24.3%) in the CG and they were omitted from a secondary analysis (data are shown in Table 2).

The comparison between the tandem and the control group revealed similar baseline PIC (29.50 ± 32.33 vs. 15.76 ± 20.93 *p* = 0.18), T_max_ > 6 s (131.45 ± 44.66 vs. 121.70 ± 73.40 *p* = 0.1), T_max_ > 10 s (69.27 ± 40.91 vs. 57.57 ± 44.47 *p* = 0.252), and FIV (54.67 ± 65.73 vs. 55.14 ± 64.64 *p* = 0.875). The presence of a target mismatch profile (and each subcomponent) was similar between the groups (data are shown in Table 3a). All patients had an absolute mismatch >15 cm^3^. Notably, mean differences between predicted and final infarct volume were not significantly different between groups (25.27 ± 46.23 vs. 39.38 ± 60.29 *p* = 0.615).

In patients who achieved a full recanalization, no significant correlation was found between mean core difference PIC to FIV and, respectively, CTP to recanalization time (tau-b = −0.15 *p* = 0.865) and procedural length (tau-b = 0.78 *p* = 0.394).

A tau-B Kendall correlation analysis showed a significant correlation between PIC and FIV both in tandem (tau = 0.743 *p* < 0.001) and in the control group (tau = 0.358 *p* = 0.003) (Figure 1).

The Bland–Altman plot showed how 95% of the differences between PIC and FIV fall in the ±2SD range with respect to the mean difference, pointing to a general agreement with the symmetric distribution of bias between the 2 groups studied (Figure 2). A linear regression analysis showed a significant relationship between PIC and FIV means and PIC and FIV differences (β = 0.761 *p* < 0.0001), stressing the presence of a positional bias in both groups.

In the secondary analysis, performed excluding patients who suffered a PH2 infarction (4 subjects in the TG and 9 subjects in the CG), there were no significant differences in demographic, clinical, radiological, interventional, and outcome features among groups as for the whole sample (data are shown in Appendix A). Patients achieving a full recanalization showed a significant lower mean in absolute core difference between PIC and FIV, with a small to medium effect size (8.74 ± 17.42 vs. 23.22 ± 32.68 U = 147.00 *p* = 0.044 r = 0.3) with respect to patients achieving a partial recanalization (data are shown in Table 4).

Again, no significant correlation between mean core difference and CTP to recanalization time (n = 31, tau-b = −0.03 *p* = 0.321) or procedural length (n = 31, tau-b = 0.09 *p* = 0.837) in patients with a full recanalization was detected (data are shown in Table 4).

A tau-B Kendall correlation analysis showed significant correlation between the baseline PIV and FIV in both the tandem (tau = 0.761 *p* < 0.001) and the control group (tau = 0.315 *p* = 0.029) (Figure 3).

The Bland–Altman plot showed how 95% of the differences between PIC and FIV fall in the ±2 SD range with respect to the mean difference, pointing to a general agreement with the symmetric distribution of bias between the 2 groups studied (Figure 4. A linear regression analysis showed a borderline significant relationship between PIC and FIV means and PIC and FIV differences (β = 0.457 *p* < 0.057) reducing the presence of the positional bias in the tandem group reported in the whole sample analysis (Figure 4).

## 4. Discussion

In AIS due to the LVO the accurate prediction of the FIV is crucial to guide the correct treatment decision making and predict functional outcome [7,8,9,10,11]. However, in spite of the availability of different techniques and progresses in mathematical models, ischemic core detection and its spatiotemporal dynamic evolution still presents several challenges in the neuroimaging field [38,39,40,41,42,43,44]. CTP is a widely accepted tool to predict FIV [9,10,11,12,13,14] in LVO by estimating CBV, CBF, or different Tmax depending on the mathematical model used by the software. Such parameters grossly rely upon the time–density curves generated by the scanning contrast bolus passage in different arterial vessels [45,46,47]. AIF location is crucial to allow the correct comparison between perfusion parameters of the two hemispheres [30,31] in the RAPID software which utilizes an automated AIF algorithm optimized for detection of candidate voxels and delay; it is corrected but uninformed about site occlusion [18,24].

In TO, the extracranial ICA occlusion is supposed to alter cerebral perfusion by bolus delay and dispersion, finally affecting main dynamic perfusion parameters such as flow or T_max_ [26,27,28,29]. Collateral status and nonlinear time-dependent evolution of ischemic core from stroke onset could generally contribute to anomalous perfusion patterns in TO [48,49].

In our study, both groups were homogenous for demographic features, clinical histories, treatment times, and outcomes (efficacy and safety of EVT) (data are shown in Table 1). As expected for the occlusion site, TG showed a higher incidence of atherothrombotic etiology, a higher median NIHSS, and a higher CBS, while from a procedural standpoint, it showed a higher number of EVT passages, longer procedural times, and a longer interval from CTP to recanalization. This last feature did not cause a disproportionate impact on FIV for TG since no significant difference was detected among groups for FIV, maybe due to the highly overlapping “onset to recanalization times” between the two groups (data are shown Table 2). Furthermore, all perfusion parameters were homogenous between the two groups (data are shown in Table 3).

The correlation analysis (Figure 1) and the Bland–Altman plot (Figure 2) showed a significant correlation and a concordance between PIC and FIV for both groups. However, as occurred in Haussen et al. and Albers et al. [9,24], a significative direct correlation between mean PIC and FIV means and mean PIC and FIV differences, visually detected as a higher dispersion of mean differences at higher core volumes (Figure 2), prevented generalization of the validity of the Bland–Altman plot [32,33]. We suppose this phenomenon could be due to an overestimation of the FIV due to the increased risk of hemorrhagic infarction after EVT, mostly in the case of greater baseline ischemic cores.

In our sample, the overall rate of post-procedural hemorrhagic transformations is 52% (50% in TG and 54% in CG), with parenchymal hematoma type 2 (PH2) representing 42% of the total (18.2% in TG and 24.3% in the CG) (data are shown in Appendix A). This is a high incidence with respect to data coming from large registries and metanalyses [50,51]. This could be a reflection of a selection bias due to inclusion criteria. In fact, high onset to treatment times (clinical indication for CTP is beyond 6 h from stroke onset or unknown time of onset), high baseline ischemic cores (8 patients were treated with ischemic core > 50 cm^3^), and a relatively high proportion of patients with TO (37%), which showed higher median NIHSS and are at relatively higher risk of post-procedural hemorrhagic transformation [52], are reported in our study. A hemorrhagic infarction, as occurs in PH2, should alter the correct evaluation of the FIV, as per the PH2 definition itself [37]. Moreover, a significant difference was found between the FIV and FIV-PIC differences between PH2 and other subtypes of hemorrhagic transformation (data and statistical analysis are reported in the Appendix A). Overestimating FIV by including PH2 cases in the analysis could justify the higher difference between PIC and FIV in the primary analysis for both groups (data are shown in Table 3a) and the unexpected similarity of the mean difference between PIC and FIV both in partial and full recanalization (data are shown in Table 3b).

In the secondary analysis, performed by excluding PH2 cases, we observed an important reduction in the mean difference between PIC and FIV: 9.33 mL ± 16.06 vs. 25.27 ± 46.23 for the tandem group and 16.11 mL ± 27.84 vs. 39.38 ± 60.29 for the control group. In the Bland–Altman plot, the differences between PIC and FIV were more equally distributed across the mean differences irrespective of PIC and FIVs means for both groups (Figure 4); moreover, considering only fully recanalized patients (mTICI 3), the whole sample mean difference between PIC and FIV lowered to 8.74 mL ± 16.06, which was very close to the median difference of 11.0 cm^3^ as recorded in the SWIFT-PRIME trial and 8 cm^3^ in Haussen et al. [24].

This study presents several limits; first of all, the retrospective observational design and the small sample size that prevented conducting a reliable analysis for the main determinants of clinical outcomes. Patients were selected for CTP perfusion not only according to the DEFUSE and DAWN criteria (>6 h from stroke onset) and the sample included some wake-up and unknown onset strokes. Accordingly, some PIC determination could be influenced by the early time window [44]. Intracranial occlusions such as T occlusion or concomitant anterior cerebral artery + MCA occlusion were not represented in the sample so as to minimize errors in FIV and PIC calculation, reducing the generalizability of the results. Despite the non-significant differences detected among TG and CG in our study, perfusion parameters such as T_max_, MTT, and CBV, whose threshold values could be affected by delay and dispersion phenomena in TO, remain somewhat elusive; this could be clinically relevant since the actual treatment criteria are based on the mismatch between the ischemic core and the penumbra tissue. Further investigations are needed to implement the reliability of CTP in predicting the pathophysiological evolution of ischemic tissue in more complicated intracranial LVOs.

## 5. Conclusions

In the case of a full recanalization and in the absence of procedural complications such as large hemorrhagic transformations, an automated CTP with rigorous normalization, thresholding, and voxel-wise analysis does not seem to be affected by concomitant extracranial ICA occlusion in predicting ischemic core in AIS due to the LVO of the anterior circulation.

## Figures and Tables

**Figure 1 jpm-13-00342-f001:**
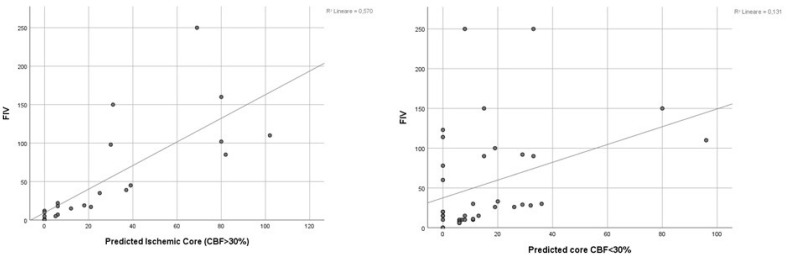
Scatter plot of PIC and FIV in the tandem group (on the **left**) and in the MCA-LVO control group (on the **right**).

**Figure 2 jpm-13-00342-f002:**
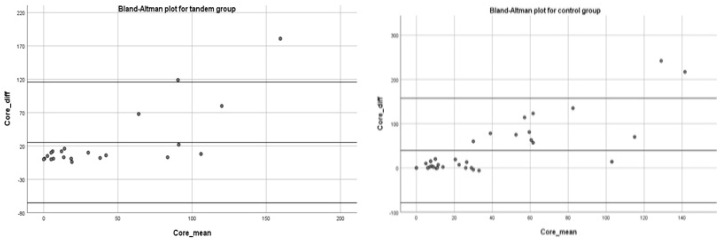
Bland–Altman plot for the tandem group (on the **left**) and the MCA–LVO control group (on the **right**).

**Figure 3 jpm-13-00342-f003:**
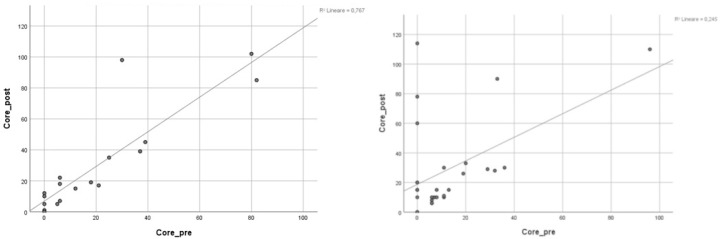
Secondary analysis scatter plot between PIC and FIV in the tandem group (on the **left**) and the MCA-LVO control group (on the **right**).

**Figure 4 jpm-13-00342-f004:**
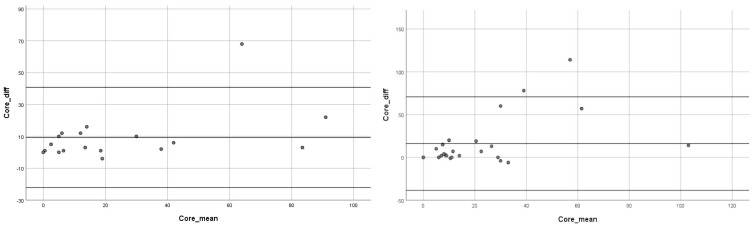
Secondary analysis Bland–Altman plot for the tandem group (on the **left**) and the MCA–LVO control group (on the **right**).

**Table 1 jpm-13-00342-t001:** Comparison between groups for baseline characteristics.

	Tandem Group (22)	Control Group (37)	*p*
Gender, n (%)			
Male	10 (45.5)	13 (34.5)	0.268
Female	12 (54.5)	24 (65.5)	0.268
Age, median (IQR ^a^)	69.5 (15)	76 (16)	0.242
Pre-stroke mRS ^b^, median (IQR)	0 (0)	0 (0)	>0.05
Arterial hypertension, n (%) *	18 (81.8)	25 (69.4)	1.00
Atrial fibrillation, n (%) *	9 (37.8)	14 (40.9)	0.852
Hypercholesterolemia (%) *	4 (18.2)	13 (36.1)	0.234
Diabetes, n (%) *	2 (9.1)	7 (19.4)	0.459
Smoking, n (%) *	3 (13.6)	4 (11.1)	0.775
Alcohol (%) *	2 (9.1%)	0	0.066
Etiology-TOAST (%)			0.055°
Undetermined	3 (13.6)	13 (35.1)	
Large artery	11 (45.5)	7 (18.9)	
Cardioembolic	7 (31.8)	16 (43.2)	
Other	2 (9.1)	1 (2.7)	
ASPECTS ^c^, median (IQR)	9 (2)	9 (3)	0.704
Good Collateral Score, n (%)	13 (72.2)	26 (59.1)	0.301
NIHSS ^d^, median (IQR)	19 (6)	15 (8)	**0.031**
Clot Burden Score, median (IQR)	3 (3)	6 (2)	**0.000**
Occlusion site, n (%)			0.247
M1 ^e^ proximal	18 (81.8)	24 (64.9)	
M1 distal	0	3 (8.1)	
M2 ^f^	4 (18.2)	10 (27)	
Occlusion side, n (%)			0.712
left	10 (45.5)	15 (40.5)	
right	12 (54.5)	22 (59.5)	
Systolic blood pressure **	139.4 (28.38)	134.71 (22.43)	0.591
Diastolic blood pressure **	86.35 (22.91)	76.34 (12.9)	**0.024**
Glycaemia *	138. 14 (44.15)	130.36 (62.63)	0.265

^a^ IQR: interquartile range; ^b^ mRS: modified Rankin Scale; ^c^ ASPECTS: Alberta Stroke Program Early CT Score; ^d^ NIHSS: National Institute of Health Stroke Scale; ^e^ M1: middle cerebral artery M1 tract; ^f^ M2: middle cerebral artery M2 tract; * Missing N = 1; ** Missing N = 2 for control group and N = 2 for tandem group.

**Table 2 jpm-13-00342-t002:** Comparisons between groups for interventional and outcome measures.

	Tandem Group (22)	Control Group (37)	*p*
Onset to groin (SD ^a^) **	269.56 (185.56)	277.04 (119.03)	0.228
Onset to CTP ^b^ (SD) **	212.38 (196.72)	214.19 (112.71)	0.157
CTP to recanalization (SD)	135.74 (48.51)	107.49 (37.93)	0.046
Onset to reperfusion, mean *** (SD)	337.31 (195.41)	321.13 (117.12)	0.662
Procedural time, mean (SD)	67.68 (47.09)	40.14 (26.95)	**0.013**
rTPA ^c^, n (%)	7 (31.8)	7 (18.9)	0.345
General anesthesia	0	2 (5.4)	0.267
Passages:			
Median (IQR ^d^)	2 (2)	1 (1)	**0.004**
Technique:			**0.02**
Direct aspiration (%)	14 (63.6)	35 (94.6)	X^2^ = 9.394
Stent retriever (%)	0	0	
Solumbra (%)	8 (36.4) *	2 (5.4)	
Successful recanalization (mTICI ^e^ 2b − 3)	14 (63.6)	25 (67.6)	0.758
Functional independence, n (%) *	9 (47.4)	16 (45.7)	0.907
Hemorrhages (%)			0.924
Total	11 (50)	20 (54.9)	
HI1 ^f^	2 (9.1)	2 (5.4)	
HI2	3 (13.1)	4 (10.8)	
PH1 ^g^	2 (9.1)	5 (13.5)	
PH2	4 (18.2)	9 (24.3)	
sICH ^h^ (%)	3 (13.6)	4 (10.8)	0.746
Unfavorable outcome, n (%)	8 (40)	15 (42.9)	0.836

^a^ SD: standard deviation; ^b^ CTP: computed tomography perfusion; ^c^ rTPA: recombinant tissue plasminogen activator ^d^ IQR: interquartile range; ^e^ mTICI: modified thrombolysis in cerebral infarction; ^f^ HI: hemorrhagic infarction; ^g^ PH: parenchymal hemorrhage; ^h^ sICH: symptomatic intracranial hemorrhage; * missing n = 3 in tandem group, n = 2 in control group; ** missing n = 6 in tandem group and 14 in control group; *** missing n = 5 in tandem group and n = 13 in control group.

**Table 3 jpm-13-00342-t003:** Comparisons between groups for perfusional parameters.

(**a**)			
	**Tandem Group (n = 22)**	**Control Group (n = 37)**	** *p* **
PIC ^a^ (SD ^b^)	29.50 (32.33)	15.76 (20.93)	0.180
FIV ^c^ * (SD)	54.67 (65.73)	55.14 (64.64)	0.875
Mean difference PIC-FIV (SD)	25.27 (46.23)	39.38 (60.29)	0.615
Core> 50 cm^3^ (%)	5 (22.7)	3 (8.1)	0.234
Tmax ^d^ 6 sec (SD)	131.45 (44.66)	121.70 (73.40)	0.100
Tmax 10 sec (SD)	69.27 (40.91)	57.57 (44.47)	0.252
Tmax > 10 > 100 cm^3^ (%)	6 (16.2)	6 (27.3)	0.308
Hypoperfusion Index	0.48 (0.22)	0.44 (0.22)	0.359
(**b**)			
	**mTICI ^e^ 2b (n = 20)**	**mTICI 3 (n = 39)**	** *p* **
Mean absolute core difference FIV—PIC	32.30 (41.11)	35.50 (30.67)	0.328

^a^ PIC: predicted ischemic core volume by CBF reduction < 30%; ^b^ SD: standard deviation; ^c^ FIV: final infarct volume; ^d^ T_max_: time to maximum; ^e^ mTICI: modified thrombolysis in cerebral infarction; * determined by MRI ≤ 48 h hours in n = 55 cases (96%).

**Table 4 jpm-13-00342-t004:** Comparisons between groups for perfusion parameters—secondary analysis.

(**a**)			
	**Tandem Group (18)**	**Control Group (28)**	** *p* **
PIC ^a^ (SD ^b^)	20.39 (25.51)	12.79 (19.72)	0.383
FIV ^c^ * (SD)	29.72 (32.65)	28.89 (31.71)	0.973
Mean difference FIV-PIC (SD)	9.33 (16.06)	16.11 (27.84)	0.821
Core > 50 cm^3^ (%)	4 (11.1)	0	0.145
T_max_ ^d^ 6 s (SD)	126.67 (47.18)	127.00 (77.08)	0.380
T_max_ 10 s (SD)	62.44 (41.73)	57.29 (46.01)	0.599
T_max_ > 10 s > 100 cm^3^ (%)	4 (22.2)	4 (14.3)	0.693
Hypoperfusion Index	0.44 (0.21)	0.41 (0.22)	0.558
(**b**)			
	**mTICI ^e^ 2b (n = 15)**	**mTICI 3 (n = 31)**	** *p* **
Mean absolute core difference FIV—PIC	23.22 (32.68)	8.74 (17.42)	**0.044**

^a^ PIC: predicted ischemic core volume by CBF reduction < 30%; ^b^ SD: standard deviation; ^c^ FIV: final infarct volume; ^d^ T_max_: time to maximum; ^e^ mTICI: modified thrombolysis in cerebral infarction; * determined by MRI ≤ 48 h hours in n = 55 cases (96%).

## Data Availability

The data presented in this study are available on request from the corresponding author. The data are not publicly available due to privacy.

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
