# Peer review of "CT Perfusion as a Predictor of the Final Infarct Volume in Patients with Tandem Occlusion"

_jpm, 2023, doi:10.3390/jpm13020342_

Round 1
Reviewer 1 Report
The idea of the study is good, but at the same time there are many problems related to the design and interpretation.
1. haemorragic – spelling
2. define PH2, CBF in abstract
3. The authors excluded PH2 = parenchymal hematoma type 2 cases from the analysis. What about HT1, HT2, PH1? The appearance of HT2, PH1 after treatment can also influence the results and conclusions. How many patients included with HT1, HT2 and PH1? Did you analyed these subgroups of patients?
4. In abstract: FIV (54,67 ± 65,73 vs 55,14 ± 64,64) - missing p value
5. In Introduction - emisphere , carothid, ematoma - spelling!!!
6. MEthods: Categorical variables are expressed as proportions (%). – N is missing.
7. In Table 1, please define which patients make up the control group.
8. In Table 1: Controls consists of 37 patients, however male N=13, female=26!!!!!!!Please correct the Table!
9. Results: Fully reperfused patient (mTICI 3) showed a mean higher difference in absolute core difference (32,30±41,11 vs 35,50 ± 30,67 p=) – missing p value
10. According to their data, the rate of post-procedural bleeding is quite high. See, Csecsei, P., Tarkanyi, G., Bosnyak, E., Szapary, L., Lenzser, G., Szolics, A., Buki, A., Hegyi, P., Abada, A., & Molnar, T. (2020). Risk analysis of post-procedural intracranial hemorrhage based on STAY ALIVE Acute Stroke Registry. Journal of stroke and cerebrovascular diseases : the official journal of National Stroke Association, 29(7), 104851. https://doi.org/10.1016/j.jstrokecerebrovasdis.2020.104851 or Hao, Y., Zhang, Z., Zhang, H., Xu, L., Ye, Z., Dai, Q., Liu, X., & Xu, G. (2017). Risk of Intracranial Hemorrhage after Endovascular Treatment for Acute Ischemic Stroke: Systematic Review and Meta-Analysis. Interventional neurology, 6(1-2), 57–64. https://doi.org/10.1159/000454721, where the frequency of ICH was much lower. How do you explain this significant difference?
11. Was attention paid to distinguishing post-procedural bleeding from hyperdensity caused by the very common exit of contrast material?
12. I see that the number of missing patient data has also been highlighted in many places. Why weren't these excluded from the anyalsis?
13. Discussion: In our study both groups were omogenous – spelling!
14. Conclusion: In the case of optimal recanalizations (TICI 3), and in the absence of procedural complications (PH2), the ability to predict the ischemic core (CBF< 30%) determined with automated software RAPID is equally reliable in tandem occlusions as in anterior circulation LVO. - With such a low number of cases, I would only make such a statement very cautiously.
15. There would be a need for a Table where the authors would present the outcome of the postprocedural ICH (HT1, HT2-PH1) cases, compared to similar cases in the Control group.
Author Response
Thank you for the time spent reading our paper (above all during Christmas holiday).
Please find the point to point reply in the attachment

Reviewer 2 Report
The authors presented a manuscript regarding usefulness of CT perfusion as predictor of the final infarct volume in patients with tandem occlusion. The manuscript is well written with novel data and results suggesting that CTP is a method of choice to evaluate the perfusion of brain parenchyma in ischemic stroke TOs patients. I suggest the authors revise the results section regarding better presentation and fewer tables used so that the section could be more concise. I thank the authors for the great topic and research field that they evaluated.
Author Response
Thank you for the time spent reading our paper (above all during Christmas holiday). Accordingly with your report we propose to cut mostly of the first part of the results, including figure 1 (recruitment chart), uploading it in supplemental material section along with the baseline characteristics of the sample in the secondary analysis (table 5 and 6) which grossly overlap with those of the full sample, so focusing mainly on the data about perfusional parameters. We've tried to clear the tables as to make it more user friendly. For the same reason we join the figures about the correlation analysis and Bland Altman plot. we're waiting to upload this new draft. We hope you could enjoy it.
Round 2
Reviewer 1 Report
The authors answered all my relevant questions, corrected the requested errors. Overall, the manuscript in this form is recommended for publication!